# Deoxynivalenol and Zearalenone: Different Mycotoxins with Different Toxic Effects in the Sertoli Cells of *Equus asinus*

**DOI:** 10.3390/cells10081898

**Published:** 2021-07-27

**Authors:** Jun-Lin Song, Guo-Liang Zhang

**Affiliations:** 1College of Animal Science and Technology, Qingdao Agricultural University, Qingdao 266109, China; 201701051@qau.edu.cn; 2Central Laboratory, Qingdao Agricultural University, Qingdao 266109, China

**Keywords:** DON, ZEA, *Equus asinus*, Sertoli cells, pyroptosis, RNA-seq

## Abstract

(1) Background: Deoxynivalenol (DON) and zearalenone (ZEA) are type B trichothecene mycotoxins that exert serious toxic effects on the reproduction of domestic animals. However, there is little information about the toxicity of mycotoxins on testis development in *Equus asinus*. This study investigated the biological effects of DON and ZEA exposure on Sertoli cells (SCs) of *Equus asinus*; (2) Methods: We administered 10 μM and 30 μM DON and ZEA to cells cultured in vitro; (3) Results: The results showed that 10 μM DON exposure remarkably changed pyroptosis-associated genes and that 30 μM ZEA exposure changed inflammation-associated genes in SCs. The mRNA expression of cancer-promoting genes was remarkably upregulated in the cells exposed to DON or 30 μM ZEA; in particular, DON and ZEA remarkably disturbed the expression of androgen and oestrogen secretion-related genes. Furthermore, quantitative RT-PCR, Western blot, and immunofluorescence analyses verified the different expression patterns of related genes in DON- and ZEA-exposed SCs; (4) Conclusions: Collectively, these results illustrated the impact of exposure to different toxins and concrete toxicity on the mRNA expression of SCs from *Equus asinus* in vitro.

## 1. Introduction

Deoxynivalenol (DON) and zearalenone (ZEA) are *Fusarium* mycotoxins produced by *Fusarium* fungi [1]. DON frequently occurs in the feed of livestock in combination with ZEA [2]. It is inevitable that grains will be contaminated by mycotoxins such as DON or ZEA during growth, storage, and processing [3,4]. Previous in vitro studies demonstrated that DON or ZEA exposure is linked to reproductive disorders in farm animals [5,6,7,8], and the toxins cause major economic loss in domestic animal industries [5,9]. In addition, ZEA causes reproductive disorders and demonstrates species-specific, organ-targeted oestrogenic activity in farm animals [10]. The effects include ovarian atrophy in pigs [11], follicular haematomas in horses [5], and reproductive failure in domestic animals [12,13,14,15]. However, the specific toxicity of DON and ZEA on reproduction in male *Equus asinus* is still unclear.

*Equus asinus* is a domestic animal that serves as a pet and draft animal and is important in mule and milk production worldwide [16,17]. According to the statistics of the Food and Agriculture Organization of the United Nations (FAO), in 2019, there were 50,583,572 *Equus asinus* in the world, of which 2,600,700 were in China (http://www.fao.org/faostat/zh/#data/QA; accessed on 7 June 2021). China has a long history of raising *Equus asinus* for agriculture and transportation [8,18,19]. Moreover, *Equus asinus* is the major source of the Chinese traditional tonic E Jiao, the production of which was worth 17.8 billion dollars in 2020. Previous in vitro investigations indicated that DON or ZEA exposure may influence the genomic stability of *Equus asinus* and porcine granulosa cells (GCs) [7,20]. In addition, horses fed oats contaminated with ZEA have a high incidence of follicular haematomas [5,21]. The aberrant development of Sertoli cells is also related to some primary testicular carcinomas in animals and humans [22,23]. Abnormal expression of functional genes in Sertoli cells such as androgen-regulated homeobox gene (NKX3.1) was seen in testicular Sertoli cell tumors [24].

According to daily testing of mycotoxins in feed from the National Black *Equus asinus* Breeding Center in China, the annual average levels of DON and ZEA were 646.90 μg/kg and 545.52 μg/kg, respectively, in 2020. However, the annual average levels of DON and ZEA were also extremely high compared to US Food and Drug Administration (FDA) mycotoxin regulatory guidance (<500 μg/kg) [25] and European Commission Guidance (EU) standards (<250 μg/kg) [26] for animal feed. In August, the contents of DON and ZEA in forage had reached their annual peak, with monthly averages of 1386.24 μg/kg and 1170.47 μg/kg, respectively, which was more than twice as high as the national standard for porcine feed (<500 μg/kg). To date, the national or local standards of mycotoxin content in the forage of *Equus asinus* have not been publicised in China. But high levels of mycotoxins in feed, especially DON and ZEA, have severely impacted the breeding of *Equus asinus*.

SCs perform numerous functions in vivo, including secretion of factors and nutrition of germ cells [27], coordinating spermatogenesis and formation of the blood–testis barrier, and phagocytosis of degenerating germ cells [28,29]. There are foetal and adult SC populations that differ in function and age in the testis [30,31]. Only immature SCs can proliferate [29,32]. Therefore, adult SCs have a finite number of germ cells that they can support [32]. Accordingly, the efficiency of spermatogenesis is influenced by SCs, and factors affecting SCs influence testis size and daily sperm production [30]. In addition, SCs are the most widely used testicular cells for in vitro toxicology [29,33,34]. The testis is highly vulnerable to xenobiotics and toxins, yet the number of studies undertaken in *Equus asinus* is insufficient and should be drastically increased.

## 2. Materials and Methods

### 2.1. Reagents

DON and ZEA were procured from Sigma-Aldrich^®^ (D0156 and Z2125 St. Louis, MO, USA). Stock solutions of DON and ZEA at a concentration of 20,000 μM were prepared by dissolving the toxins in dimethyl sulfoxide (DMSO) and stored at −80 °C. DMSO (D12345), M-199 medium (11150067), foetal bovine serum (FBS, 10091148), penicillin, and streptomycin were provided by Thermo Fisher Scientific (Waltham, MA, USA). Collagenase Ⅳ (C8160), hyaluronidase (H8030), and DNase Ⅰ (D8071) were provided by Solarbio (Beijing, China). Trypsin-EDTA solution (0.25%, SH30042.01) was provided by HyClone (Beijing, China).

### 2.2. Animals

Prepubertal *Equus asinus* testes were collected from the slaughterhouse of Dong E *Equus asinus* Production Company (Qingdao, China) and maintained at 32 °C–35 °C Hank’s balanced salt solution (Sigma-Aldrich, St. Louis, MO, USA) for the isolation of SCs. All *Equus asinus* used in the experiment were fed and treated humanely during slaughter (no. 11002009000012, production license number: SCXK: 2020-7077, Qingdao, China).

All animal experiments were performed in accordance with Chinese welfare guidelines and approved by the Institutional Animal Ethical Committee of Qingdao Agricultural University (DEC 2020-019). All experiments were performed in accordance with relevant guidelines and regulations.

### 2.3. Dose Selection

The annual average dose of DON and ZEA from the “National Black *Equus asinus* Breeding Center” in 2020 was selected and translated to SCs in vitro. The doses of toxins were standardised according to the results of previous studies on ZEA-induced impairments in the fertility of *Equus asinus*, pigs, and mice [8,35]. DMSO was used as the vehicle for DON and ZEA. For accuracy, the same dose of DMSO added to the 10 μM and 30 μM DON or ZEA groups was added to the control group. Culture media were supplemented with DMSO at a final concentration of 0.1%.

### 2.4. Isolation and Culture of Equus asinus SCs

*Equus asinus* testes were collected and transported to the laboratory at 37 °C with 10% (*v/v*) antibiotic-antimycotic solution (Sangon Biotech, Shanghai, China) for 30 min. The testes were longitudinally split, and the tissue was fragmented into fine pieces by surgical scissors and digested using 1.5 mg/mL collagenase IV and 0.2% hyaluronidase mixed solution (Sigma-Aldrich, St. Louis, MO, USA) in HBSS at 37 °C for 30 min. Then, 10 μg/mL DNase I and 0.25% trypsin were used to digest the mixed solution at 37 °C for 15 min. The digested cells were filtered through 74 μm nylon mesh (Sangon Biotech, Shanghai, China). A sedimentation step collected cell clumps at the bottom of a tube after separating the filtered cells under unit gravity by incubating in a 15 mL tube (Corning, NY, USA) at room temperature for 10 min. This procedure was repeated twice to remove debris included in the cell clump. Subsequently, an adherent step separated the attached SCs and suspended cells by incubating the clumped cells on a 10 cm culture plate (Corning, NY, USA) at 37 °C for 20 min, which was repeated three times. Finally, isolated SCs were identified using immunohistochemistry methods. The fibroblast used for the contrast group being cultured from the cell clump of *Equus asinus* testes.

### 2.5. Flow Cytometry Analysis of Apoptosis

SCs were treated with DON and ZEA at 10 μM and 30 μM for 72 h. Then, they were collected and washed three times using PBS. An Annexin V-FITC/7-AAD kit (Sino Biological, APK10448-F, Beijing, China) was used to detect apoptosis or necrosis of *Equus asinus* SCs by flow cytometry according to the manufacturer’s instructions.

### 2.6. Immunofluorescence and Cell Counting

Cultured SCs were fixed in 4% paraformaldehyde for 2 h, heated at 42 °C for another 2 h, and fixed on polysine-coated slides. SC sections were blocked with 10% goat serum for 40 min and incubated with primary antibodies overnight at 4 °C (Table 1). Subsequently, the sections were incubated for 1.5 h at 37 °C with Alexa Fluor 488/Alexa Fluor 594-goat anti-rabbit secondary antibody at a dilution of 1:100 (ABclonal, AS053/AS039, Wuhan, China). The sections were incubated for 5 min with Hoechst 33,342 (Solarbio, C0031) to visualise the nuclei of SCs. Immune signals were detected using a fluorescence microscope (Olympus, XB51, Japan), and captured images were analysed in accordance with the ImageJ standard.

### 2.7. Western Blot Analysis

Proteins isolated from SCs were used for Western blot analysis in accordance with previous standard methods [36,37]. Proteins from SCs in each treatment group were separated through 10% SDS-PAGE and transferred to PVDF membranes. The membranes were incubated overnight at 4 °C with primary antibodies (Table 1), rinsed three times with TBST, and incubated for 2 h at 37 °C with secondary antibodies (Sangon Biotech, D110058) in TBST. Related proteins were detected using AlphaImager^®^ (ProteinSimple, 92-13824-00, San Jose, CA, USA) HP. The intensity of all bands was quantified with *GAPDH* as the internal control using ImageJ software.

### 2.8. TUNEL Staining

The apoptosis rates of SCs were evaluated using a TUNEL BrightRed Apoptosis Detection Kit (Vazyme, A11302, Nanjing, China). Briefly, SCs were fixed for 2 h with 4% paraformaldehyde after 72 h of exposure to 0, 10, or 30 µM DON and ZEA. After the TUNEL reaction, the cells were observed under fluorescence microscopy in accordance with the manufacturer’s instructions. TUNEL-positive cells were detected and counted under fluorescence microscopy (Olympus, XB51, Tokyo, Japan). More than 2000 SCs were obtained from each group and counted. Three biological replicates were used for analysis of the TUNEL-positive cell ratio.

### 2.9. RNA Extraction, Reverse Transcription, and RNA-Seq

RNAex pro reagent (AG, AG21101, Beijing, China) was used to extract total RNA from cultured DON- and ZEA-exposed SCs in accordance with the manufacturer’s instructions. Then, mRNA was reverse transcribed into first-strand cDNA with a cDNA synthesis kit (TransGen, AT311-03, Beijing, China), referring to a previous study [35,38]. The Novogene Company performed RNA sequencing with the 4000 platform (Beijing, China).

### 2.10. Identification of Differentially Expressed Genes

The NovoMagic and R Bioconductor/DESeq2 packages were used to identify differentially expressed genes (DEGs) between different groups of SCs (0, 10, and 30 μM DON and ZEA treatment groups). Raw counts for differential expression analysis were obtained by using NovoMagic online and checked by using our own normalisation approach [35,39]. Data for differential expression analysis were previously normalised through other methods to prevent possible biases [40,41]. The log2|fold change|was disallowed as the filter condition because the sequencing design contained biological replicates for each group. Adjusted *p* < 0.01 was considered statistically significant.

### 2.11. GO and KEGG Enrichment Analysis

The functional profiles of DEGs were analysed through GO functional enrichment and KEGG pathway analysis by using NovoMagic. NovoMagic is the analysis platform developed by Novogene Company that can visualise GO and KEGG analysis results for DEGs online. In GO analysis, genes can be categorised as molecular function, biological process, and cellular component. NovoMagic was applied to visualise the KEGG analysis results. The log2|fold change|value of DEGs reflects the active status of enriched signalling pathways. Adjusted *p* < 0.05 was considered statistically significant.

### 2.12. Quantitative Real-Time PCR

Total RNA extraction and cDNA reverse transcription were performed as previously described. A SYBR^®^ Green Premix Pro Taq HS qPCR Kit was used to perform quantitative real-time PCR (RT-qPCR) on a LightCycler^®^ 96 RT-PCR instrument (Roche, Germany). RT-qPCR was performed under the following cycling conditions: 30 s at 95 °C; 40 cycles at 95 °C (5 s), 60 °C (30 s), and 72 °C (30 s); melting at 95 °C (1 s), 65 °C (15 s), and 95 °C (1 s); and a final cooling step at 4 °C. The RT-qPCR primers used in this study are listed in Table 2. *GAPDH* was used as the reference gene for the normalisation of mRNA expression in SCs. Gene expression was quantified through the 2−∆∆CT method. The expression level of each gene is expressed as the mean ± standard deviation (SD), which was calculated from the data of at least three independent biological replicates.

### 2.13. Scanning Electron Microscopy (SEM)

The treated SCs were washed with PBS 3 times, centrifuged for 10 min at 3000 rpm at 4 °C and fixed with 2.5% glutaraldehyde overnight. Next, tert-butanol was used to separate the glutaraldehyde. After being air-dried, the slides were critical-point dried, mounted on stubs, sputter-coated with a thin layer of conductive metal, gold, and palladium, and viewed by SEM (Hitachi HT7700, Hitachi, Ltd., Tokyo, Japan).

### 2.14. Statistical Methods

Data are presented as the mean ± SD. The statistical significance of different effects among the control, DON and ZEA exposure groups of SCs was determined through one-way ANOVA for multiple comparisons. All analyses were conducted using GraphPad Prism analysis software (San Diego, CA, USA). All experiments were repeated at least three times, and the results were considered significant at *p* < 0.05.

## 3. Results

### 3.1. Apoptosis Rates and DEGs of Equus asinus SCs Exposed to DON and ZEA

First, we examined the purity of isolated SCs. The isolated cells were identified as *SOX9* (a specific Sertoli cell marker)-positive [42] using immunohistochemistry methods (purity of isolated SCs > 97%) (Figure 1A,B). Then, we exposed the SCs to 10 or 30 μM DON and ZEA for 72 h of in vitro culture (Figure 1C). As shown in Figure 2, flow cytometry analysis was used to investigate the effects of DON and ZEA on cell apoptosis (Figure 2A). The results showed that the apoptosis rate was significantly increased under DON and ZEA treatment (10 μM DON: 16.60% ± 1.39%; 30 μM DON: 22.05% ± 1.11%; 10 μM ZEA: 8.06% ± 0.49%; 30 μM ZEA: 14.58% ± 1.42%) relative to that under the control treatment (0 μΜ DON: 4.47% ± 0.24%; 0 μΜ ZEA: 4.72% ± 0.31%; *p* < 0.05 or *p* < 0.01; Figure 2B,C). Interestingly, 10 μM DON exposure remarkably increased the apoptosis rate of SCs compared with 30 μM ZEA treatment. From Appendix A, the percentages of TUNEL-positive SCs also remarkably increased under DON and ZEA treatment (10 μM DON: 22.72% ± 1.79%; 30 μM DON: 64.15% ± 3.15%; 10 μM ZEA: 15.36% ± 2.79%; 30 μM ZEA: 24.17% ± 2.22%) relative to those under the control treatment (0 μΜ DON: 3.07% ± 0.14%; 0 μΜ ZEA: 3.32% ± 0.61%; *p* < 0.01; Appendix A).

Nine libraries from the three groups were sequenced, and 715,715,682 raw reads (GEO accession number: GSE172037), with 703,892,136 clean reads, were obtained. Then, we performed RNA-seq analysis to confirm the effects of DON and ZEA exposure on SCs (Figure 3). We screened a total of 9393 and 6065 DEGs in the DON and ZEA treatment groups, respectively, based on the research criterion FDR < 0.05 (Figure 3F). We found that 3300 and 3251 genes were up- and downregulated under 10 μM DON treatment, while 4841 and 3764 genes were up- and downregulated under 30 μM DON treatment, respectively (Figure 3A,B). Furthermore, we identified 2816 and 3131 DEGs that were up- and downregulated under 30 μM ZEA treatment, while only 391 and 412 DEGs were up- and downregulated under 10 μM ZEA treatment, respectively (Figure 3C,D). Meanwhile, we selected some DEGs between the mycotoxin and control groups with degrees greater than 20 to form a heat map (Figure 3E).

We annotated the functional interactions of genes that were differentially expressed between the control and DON or ZEA treatment groups by using the STRING database to investigate the potential effects of mycotoxin exposure on SCs. Search Tool for the Retrieval of Interacting Genes/Proteins (STRING, https://string-db.org/; accessed on 7 June 2021) is a database of protein-protein interaction. This database contains the direct and physically related interactions between known and predicted protein and genes. The R Bioconductor/STRINGdb was applied for PPI of interested DEGs [43].

### 3.2. DEGs Involved in GO Classification and KEGG Pathways

We applied the NovoMagic and R packages to annotate the GO enrichment functions of DEGs in each group (Figure 4A–C). We found that DEGs in DNA metabolic process, immune system process, DNA repair, and apoptotic process (Figure 4A) were remarkably enriched in SCs exposed to 10 μM and 30 μM DON. Meanwhile, DEGs involved in the small molecular metabolic process, immune system process, DNA metabolic process, and cell cycle were remarkably enriched in SCs exposed to 10 μM and 30 μM ZEA (Figure 4B). Moreover, DEGs were significantly enriched in the regulation of apoptotic processes in SCs exposed to 30 μM ZEA (Figure 4C). In addition, upregulated DEGs in SCs under 10 μM DON treatment were significantly enriched in the immune system and apoptotic processes (*p* < 0.001). Downregulated DEGs in SCs exposed to 10 μM ZEA were significantly enriched in the steroid biosynthesis process [8].

We used NovoMagic and the clusterProfiler R package to identify extremely affected KEGG pathways to obtain insight into the function of DEGs (Figure 4D–F). DEGs in SCs exposed to 10 μM and 30 μM DON were significantly enriched in the PI3K/AKT [35], MAPK, and TNF signalling pathways (Figure 4D). Meanwhile, DEGs in SCs treated with 10 μM and 30 μM ZEA were significantly enriched in the regulation of the cell cycle and P53 signalling pathway (Figure 4E). In addition, we identified DEGs that were significantly enriched in the PI3K/AKT [35], MAPK, and Hippo signalling pathways after 30 μM ZEA treatment (Figure 4F).

The results of GO and KEGG pathway analyses revealed that *POLD1*, *Caspase1*, *GSDMD*, *CCL17*, and *PRDX4*, which are involved in DNA metabolism, pyroptosis, and inflammation processes, were differentially expressed in SCs exposed to 10 μM and 30 μM DON. The *CDK1*, *CCNB2*, *ESR1*, and *NOX1* genes involved in the cell cycle and steroid-related signalling pathways were changed in SCs exposed to 30 μM ZEA. *WNT2*, *MSH6*, *RAF*, and Cyclin D1 (*CCND1*), genes involved in cancer processes, were also differentially expressed after exposure to 10 μM and 30 μM DON.

### 3.3. Cellular and Molecular Effects of DON and ZEA Exposure on SCs

Exposure to 10 and 30 μM DON may lead to the pyroptosis of SCs (Figure 5, Figure 6 and Figure 7), and DON and ZEA might induce inflammation and endocrine effects in SCs through different molecular mechanisms (Figure 8, Figure 9, Figure 10 and Figure 11).

As shown in Figure 5, the number of immunofluorescence-positive genes, such as *Caspase1* (Figure 5A–C) and *GSDMD* (Figure 5D–F), remarkably increased in the 10 μM and 30 μM DON groups, whereas the genes showed no significant differences in SCs treated with 10 μM and 30 μM ZEA relative to those in the control group. Moreover, exposure to 10 μM and 30 μM DON significantly upregulated the mRNA abundance and protein levels of *Caspase1* (Figure 6A–C) in SCs. In addition, the mRNA abundance and protein levels of *GSDMD* (Figure 6B) and *GSDMD-N* (Figure 6D) were increased after the treatment of 10 μM and 30 μM DON. Representative SEM images indicated that SCs treated with 10 μM and 30 μM DON undergo membrane perforation and produce apoptotic body-like cell protrusions prior to plasma membrane rupture [44] (Figure 7). However, there were no significant differences in SCs treated with 10 μM ZEA compared with the control group (Figure 7).

Immunohistochemical results of SCs exposed to the mycotoxin indicated that the number of *CCL17*-positive cells remarkably increased in the 10 μM and 30 μM DON exposure groups (Figure 8A–C), while there was a significantly increased number of *IL10RA*-positive cells in the 10 μM and 30 μM ZEA treatment groups (Figure 8D–F). The mRNA abundance and protein levels of *CCL17* and *IL10RA* were significantly upregulated in SCs exposed to 10 μM and 30 μM DON (*p* < 0.05 or *p* < 0.01; Figure 9A,C) and ZEA (*p* < 0.05 or *p* < 0.01; Figure 9B,D) relative to those in SCs under the control treatment.

As shown in Figure 10, the number of immunofluorescence-positive genes, such as *AR*, remarkably decreased in the 10 μM and 30 μM DON exposure groups (Figure 10A–C), whereas the expression of *ESR1* genes was significantly increased in SCs treated with 10 μM and 30 μM ZEA (Figure 10D–F) relative to those in the control group. Moreover, exposure to 10 μM and 30 μM DON significantly downregulated the mRNA abundance and protein levels of the *AR* gene (Figure 11A,C), while ZEA exposure significantly upregulated the expression of the *ESR1* gene (Figure 11B–D) in SCs.

The results of the 10 μM DON and 30 μM ZEA analyses showed that DEGs were remarkably enriched in the oxidation-reduction process, PI3K-AKT signalling pathway and Ras signalling pathway (Figure 12A,D,G). Exposure to 10 μM and 30 μM DON (Figure 12B,E) and ZEA (Figure 12C,F) significantly downregulated the mRNA abundance and protein levels of *PRDX4* genes in SCs, respectively. The number of *PRDX4*-positive cells was remarkably reduced in the 10 μM and 30 μM DON exposure groups (Figure 13A–C), while there was a significantly decreased number of *PRDX4*-positive cells in the 30 μM ZEA treatment group (Figure 13D–F). In addition, the number of *NOX1*-positive cells remarkably increased in the 10 μM and 30 μM DON (Appendix A) and ZEA (Appendix A) exposure groups.

Our RT-qPCR and Western blot results indicated that SCs under 10 μM and 30 μM DON treatment exhibited significantly lower *MSH6* mRNA and protein levels than those under the control treatment (*p* < 0.05 or *p* < 0.01) (Figure 14A,D), whereas the *CDK1* and *CCNB2* genes were significantly upregulated in the 10 μM and 30 μM ZEA exposure groups (Figure 14B,C,E,F). We conducted more RT-qPCR to evaluate the expression of different transcripts in the pathways of SCs among the control, DON and ZEA treatments (Appendix A).

## 4. Discussion

*Fusarium* mycotoxins have been implicated in poor reproductive performance in domestic animals, including male *Equus asinus* [5,7,8,32,39,45,46,47]. In vitro reports with DON or ZEA demonstrated that mycotoxins are able to directly affect the reproductive [47,48,49,50], endocrine [51,52], and immune systems [53,54,55], as well as inheritance [32,45]. Previous research demonstrated that the function of SCs is essential in the processes of normal spermatogenesis and testis development [30,31,32]. Moreover, present findings suggested that individual or mixtures of *Fusarium* toxins had cytotoxic effects on porcine Sertoli and Leydig cells [56,57]. However, additive effects were not always observed for the mixtures of *Fusarium* toxins [56,58,59,60]. The present study was designed to investigate the effects of mycotoxin DON and ZEA treatment on pyroptosis, viability, the cell cycle, cell secretion, and cell inflammation in cultured SCs. We used immature SCs cultured in vitro and RNA-seq methods to compare the toxic effects of DON and ZEA to *Equus asinus* SCs. This is the first study to describe the differences in the transcriptomes of SCs between DON and ZEA exposure. Our results provide a basic database for mycotoxins in SC studies of *Equus asinus*.

Pyroptosis is a newly discovered type of regulated necrotic cell death induced by inflammasomes, such as *Caspase1* and *Caspase11* (in mouse cells) [61,62]. Both *Caspase1* and *Caspase11* can induce pyroptosis by processing Gasdermin D (*GSDMD*), yielding an N-terminal fragment that forms pores on the plasma membrane, leading to cell death [63,64]. Unlike apoptosis, pyroptosis is a highly specific type of inflammation that has been proven to be strongly associated with cancer [65]. Our results showed that the mRNA and protein expression levels of *Caspase1* and *GSDMD*-N in *Equus asinus* SCs remarkably increased upon exposure to 10 μM DON. Moreover, representative scanning electron microscopy images indicated that SCs treated with DON undergo membrane perforation and produce apoptotic body-like cell protrusions prior to plasma membrane rupture [44]. Some SCs also showed initial changes in the substructure of the plasma membrane characterised by focal disappearance of membrane structure and partial loss of continuity, as described for necrotic cells [66]. These results indicated that the canonical pyroptosis pathways in SCs had been activated and that SCs possessed the typical pyroptosis appearance [67] and severe cellular inflammation.

Furthermore, the mRNA abundance and protein levels of *CCL17* in the SCs remarkably increased after exposure to DON. High levels of *CCL17* have been found in SCs in seminoma tumours [68]. Therefore, DON treatment may promote the expression of the hallmarks of tumour formation given its effect on pyroptosis-related genes and *CCL17* expression. ZEA is a non-steroidal oestrogenic mycotoxin. In contrast, depending on the molecular structure, ZEA may have binding affinities to oestrogen receptors and, therefore, mimic oestrogenic effects in SCs. We found that exposure to 30 μM ZEA significantly increased the mRNA abundance and protein levels of *ESR1*, *ESR2*, and *IL10RA* in SCs. This phenomenon indicates that exposure to ZEA may lead to inflammation and affect the endocrine function of SCs. A total of 6551 and 803 genes were differentially expressed in SCs under the control of 10 μM DON and ZEA treatment. Furthermore, we identified 8605 and 5947 DEGs in the SCs under 30 μM DON and ZEA treatment, respectively. In summary, the above results suggest that DON treatment may lead to pyroptosis and promote the expression of oncogenes, while exposure to ZEA results in inflammation and affects endocrine disruption in cells. Furthermore, *Equus asinus* SCs are more sensitive to DON exposure than SCs treated with the same dose of ZEA.

In addition, bioinformatics analyses found that *PRDX4* and cell oxidation-reduction related genes were influenced in SCs exposed to 10 μM DON and 30 μM ZEA. Our results indicated that DON might have stronger oxidative toxicity than ZEA in *Equus asinus* SCs. Moreover, several studies have demonstrated that *MSH6* (an important component of the mismatch repair system) is a tumour-related factor that can affect tumorigenesis, proliferation, migration, and invasion effects [69]. It is well known that suppression of *MSH6* is associated with a variety of tumours [70,71]. Interestingly, it was found in this study that 10 μM DON exposure significantly decreased the mRNA and protein levels of *MSH6* in SCs. Therefore, exposure to 10 μM DON may be potentially mutagenic and carcinogenic. Several studies have indicated that *CCNB2* is a regulatory protein involved in mitosis, and its product can combine with *CDK1* to form a maturation-promoting factor [72,73,74,75]. *CCNB2* overexpression can result in uncontrolled cell growth [76,77]. 

In this study, we found that exposure to 30 μM ZEA remarkably upregulated the mRNA and protein abundance of *CCNB2* in SCs. In contrast, exposure to 10 μM ZEA resulted in *CDK1* overexpression in SCs. These results suggest that 30 μM ZEA treatment may promote the expression of oncogenes in SCs. Furthermore, exposure to 10 μM DON resulted in *IL32* and *NOX1* gene overexpression, while the *AR*, *AIG1*, *MCM6*, and *POLD1* genes were suppressed in SCs. In addition, 30 μM ZEA treatment led to *PFKM*, *NOX1*, and *ESR2* gene overexpression in the cells. In our experiments, we observed a suppressive effect on the mRNA abundance of the *AR* gene of SCs at concentrations of 10 μM and 30 μM DON treatment, while there was a stimulatory effect on the *ESR1* and *ESR2* genes of SCs under 30 μM ZEA exposure. The above results validated that DON and ZEA had differential toxic patterns in *Equus asinus* SCs. Additionally, the exact mechanisms by which DON or ZEA changes cell secretion in *Equus asinus* SCs remain the subject of further investigation.

We also found proapoptotic effects of DON and ZEA using flow cytometry and TUNEL-positive analysis. A significant reduction in the SCs was observed at toxic concentrations of both mycotoxins. The results demonstrate that the mycotoxins DON and ZEA can stimulate cell apoptosis in *Equus asinus* SCs. Both mycotoxins seem to transmit their molecular effects by influencing the MAPK signalling cascades and the protein kinase Akt, which could result in translation anomalies. However, it can be assumed that DON and ZEA modulate the process of translation at different molecular levels. Whereas DON mainly had an impact on pyroptosis and androgen disruption and therefore on the biological activity of the *Caspase1*, *GSDMD*, *CCL17*, *AR*, and *AIG1* genes, ZEA increased the abundance of *IL 10 RA*, *CDK1*, *PFKM*, *NOX1*, and specific bands of the *ESR1* and *ESR2* genes. Thus, DON and ZEA exerted toxic effects on SCs in a different manner. Further investigations are required to obtain more information about specific signalling cascades that transmit the toxicity of DON and ZEA in *Equus asinus* SCs.

## Figures and Tables

**Figure 1 cells-10-01898-f001:**
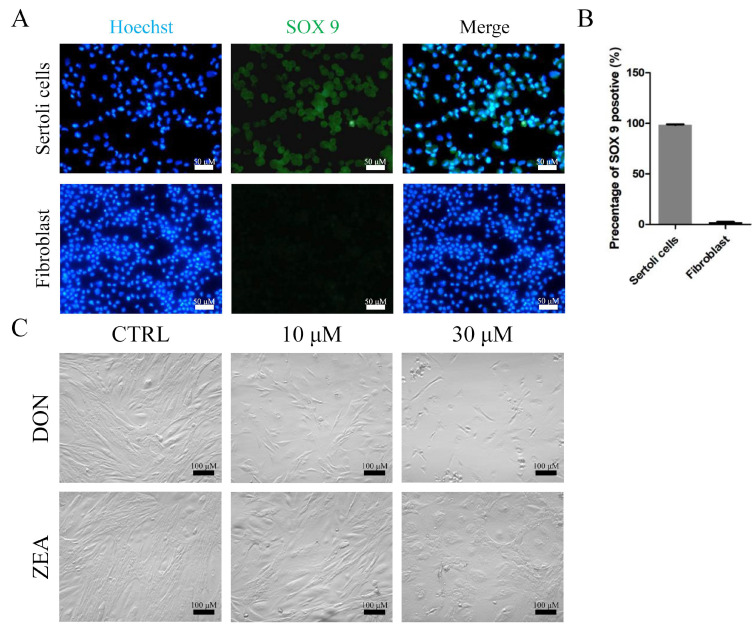
The purity of isolated SCs, as well as DON and ZEA exposure, affect the morphology of *Equus asinus* SCs. (**A**) The isolated cells were identified by *SOX9* positivity using immunohistochemistry methods, and fibroblasts were used for contrast. Bar indicates 50 μm. (**B**) The percentages of *SOX9*-positive SCs. (**C**) The morphology of SCs following administration of 0 μM, 10 μM, and 30 μM DON and ZEA after 72 h of culture in vitro. Bar indicates 100 μm.

**Figure 2 cells-10-01898-f002:**
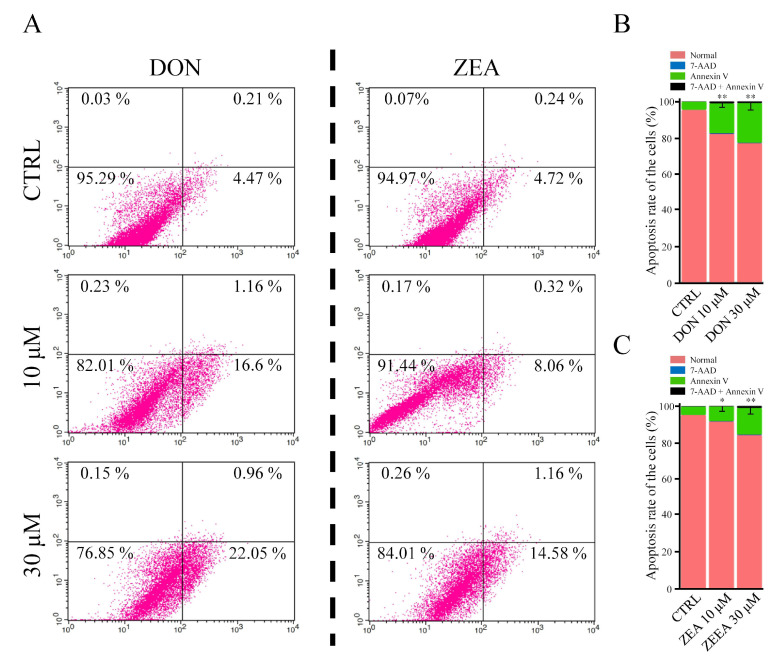
The effect of DON and ZEA on SCs apoptosis, analysed by flow cytometry. (**A**) The apoptosis results of SCs following administration of 0 μM, 10 μM, and 30 μM DON and ZEA after 72 h of culture in vitro. (**B**) The apoptosis rate of SCs after 0 μM, 10 μM, and 30 μM DON treatment. ** *p* < 0.01. (**C**) The apoptosis rates of SCs after exposure to 0 μM, 10 μM, and 30 μM ZEA treatment. * *p* < 0.05; ** *p* < 0.01.

**Figure 3 cells-10-01898-f003:**
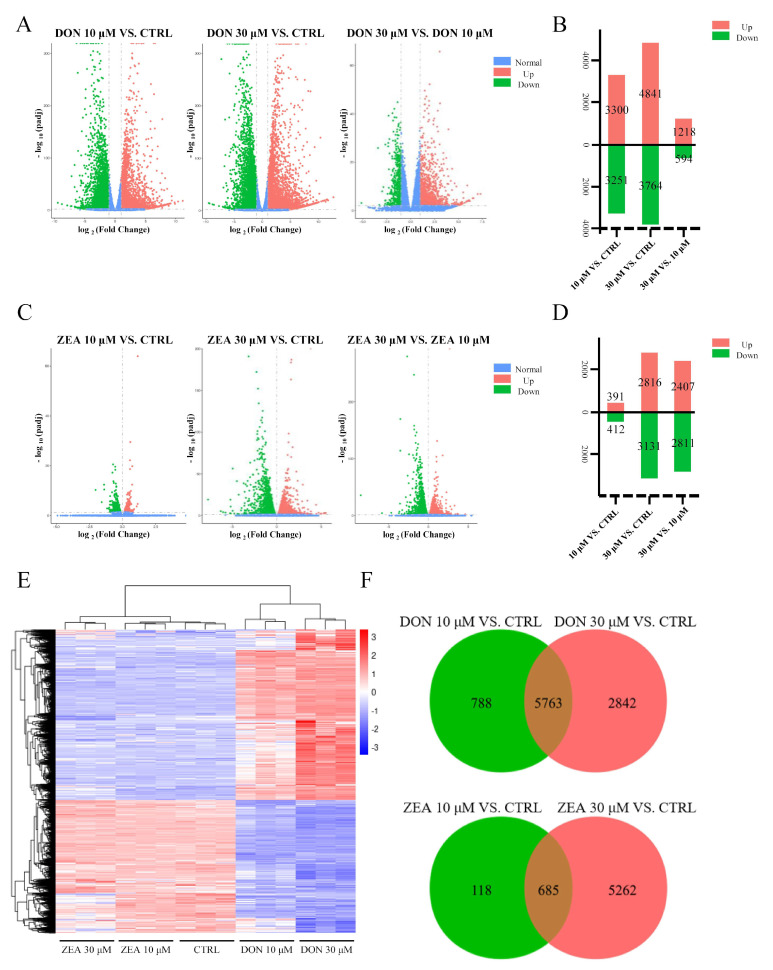
Gene expression profiles of SCs after DON and ZEA treatment. (**A**) Scatterplot of gene expression after DON treatment; 10 µM DON treatment group vs. Control group, 30 µM DON treatment group vs. Control group, 30 µM DON treatment group vs. 10 µM DON treatment group. Green and red plots represent differentially expressed genes. (**B**) Comparison of DEGs among the control, 10 µM and 30 µM DON-treated groups. (**C**) Scatterplot of gene expression after ZEA treatment; 10 µM ZEA treatment group vs. control group, 30 µM ZEA treatment group vs. control group, 30 µM ZEA treatment group vs. 10 µM ZEA treatment group. Green and red plots represent differentially expressed genes. (**D**) Comparison of DEGs among the control, 10 µM and 30 µM ZEA treated groups. (**E**) Heatmap indicating the group difference of DEGs in the 10 µM and 30 µM DON and ZEA treated groups compared with the control group, and the repeatability within each group. The results are presented as the means ± SD. All experiments were repeated at least three times. (**F**) Venn diagram showing the differential expression of 9393 and 6065 genes in the DON and ZEA treated groups compared to the control groups.

**Figure 4 cells-10-01898-f004:**
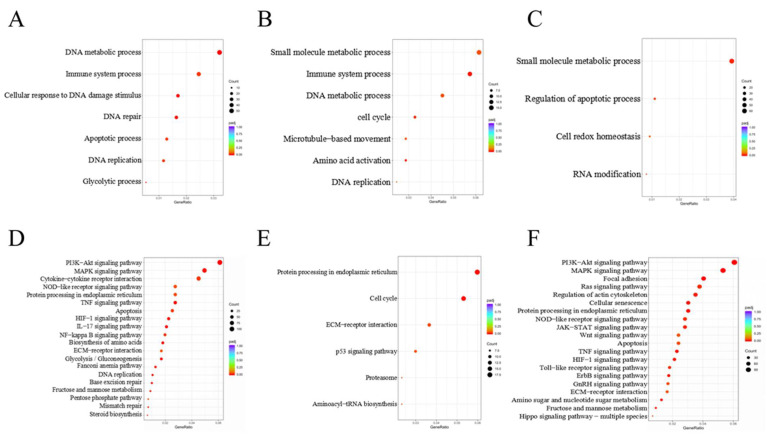
GO and KEGG biological processes involving DEGs in *Equus asinus* SCs exposed to DON and ZEA. (**A**) GO enrichment of 5763 DEGs in the 10 µM and 30 μM DON-exposed *Equus asinus* SCs, including biological process, cellular component, and molecular function. (**B**) GO enrichment of 685 DEGs in the 10 µM and 30 μM ZEA-treated *Equus asinus* SCs. (**C**) GO enrichment of 5262 DEGs in the SCs of 30 μM ZEA treatment group. (**D**) KEGG enrichment of the 5763 DEGs in the 10 μM and 30 μM DON-exposed *Equus asinus* SCs. (**E**) KEGG enrichment of the 685 DEGs in the 10 μM and 30 μM ZEA treated group. (**F**) KEGG enrichment of the 5262 DEGs of 30 μM ZEA treatment group. The lengths of the bars indicate the number of DEGs that are enriched in each pathway. The blue to red gradient represents the p.adjust trend from high to low.

**Figure 5 cells-10-01898-f005:**
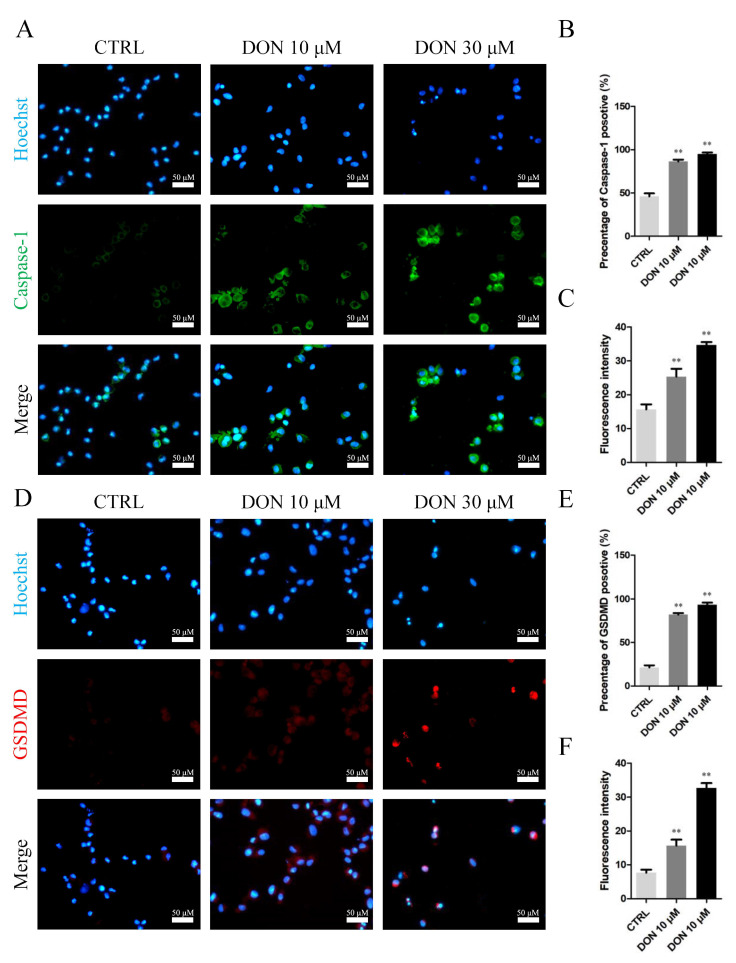
Immunofluorescence assay probing the expression of SC phosphor-*Caspase1* (**A**) and *GSDMD* (**D**) proteins in the DON treatment groups. The percentages of positive cells (**B**,**E**) and fluorescence intensity (**C**,**F**) were analysed. Bar indicates 50 μm. Data are presented as the means ± SD. ** *p* < 0.01.

**Figure 6 cells-10-01898-f006:**
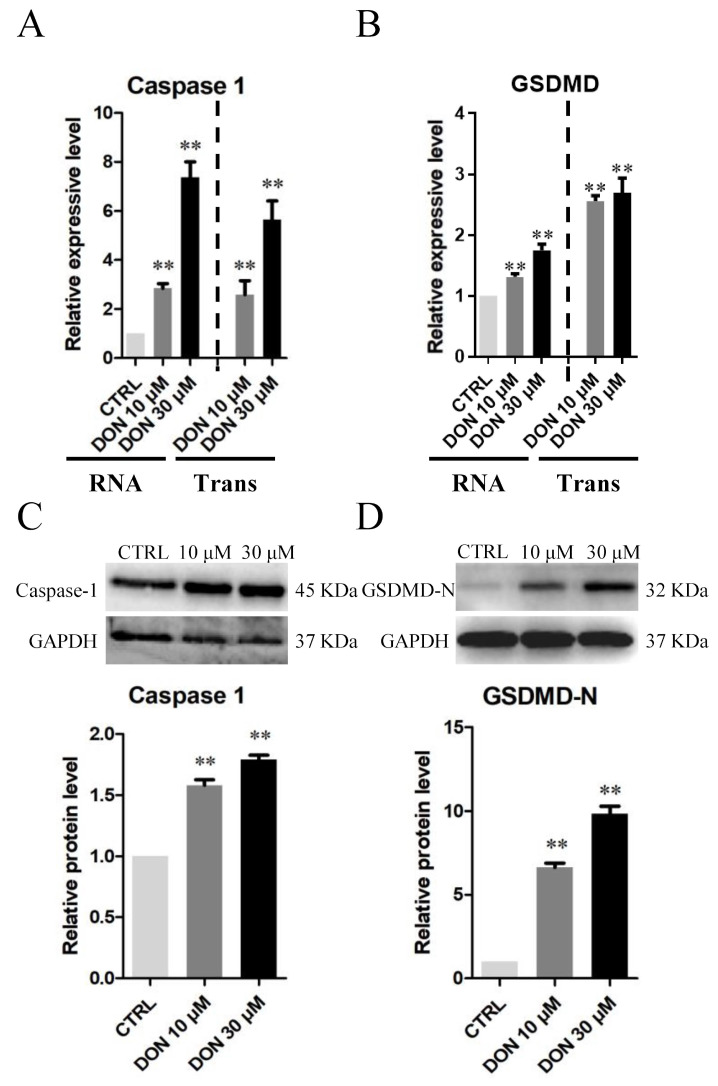
DON exposure affects the mRNA and protein abundance levels of pyroptosis-related genes in cultured *Equus asinus* SCs. (**A**) Quantitative RT-PCR for *Caspase1* transcription factors. The mRNA levels of the genes were normalised to the *GAPDH* gene. (**B**) Quantitative RT-PCR for *GSDMD* transcription factors. The mRNA levels of the genes were normalised to the *GAPDH* gene. (**C**) Protein levels of *Caspase1*/*GAPDH* by Western blot. (**D**) Protein levels of *GSDMD*-N/*GAPDH* by Western blot. The protein levels were normalised to *GAPDH*. The exposure time was 50 s. The results are presented as the means ± SD. All experiments were repeated at least three times. ** *p* < 0.01.

**Figure 7 cells-10-01898-f007:**
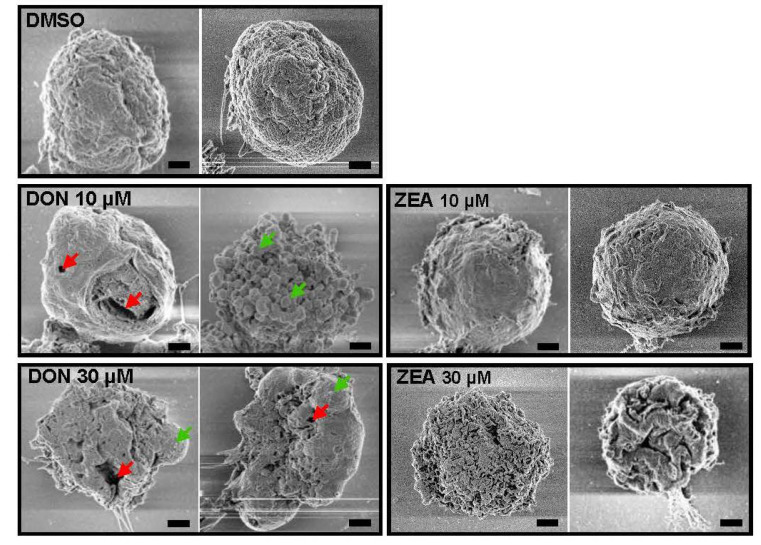
Representative scanning electron microscopy images of SCs treated with 10 μM and 30 μM DON and ZEA. Red arrows indicate membrane perforation, and green arrows indicate the apoptotic body-like cell protrusions. Bar indicates 1.0 μm in the DMSO, 10 μM and 30 μM ZEA groups. Bar indicates 1.5 μm in the 10 μM and 30 μM DON groups.

**Figure 8 cells-10-01898-f008:**
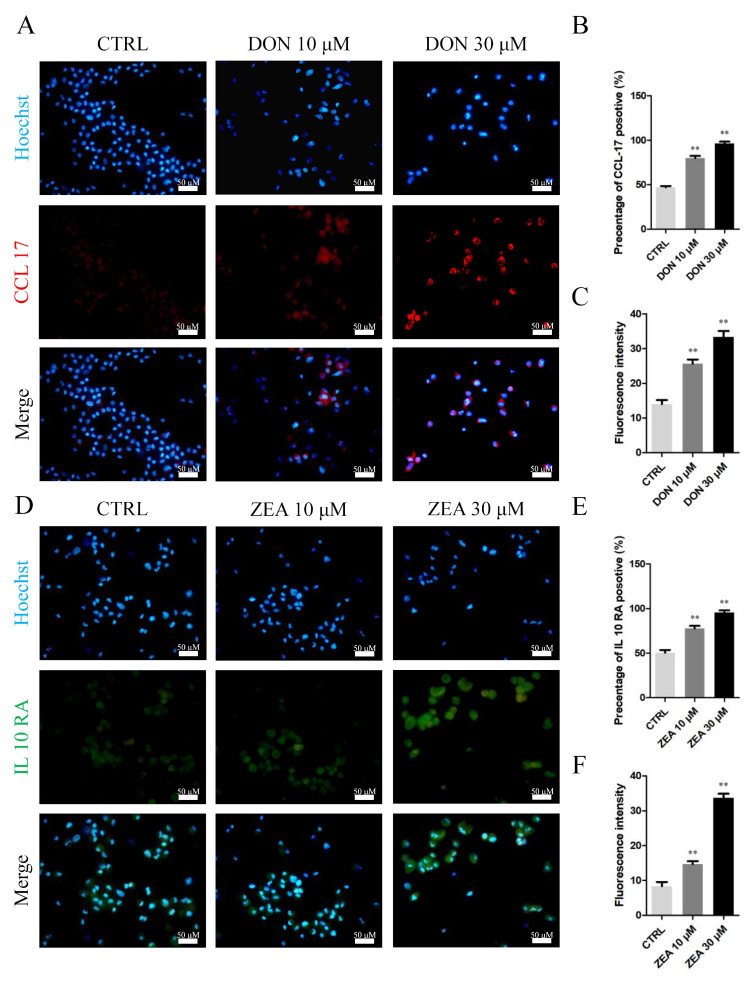
Immunofluorescence assay probing the expression of SC phosphor-*CCL17* (**A**), and *IL10RA* (**D**) proteins in the DON and ZEA treatment groups. The percentages of positive cells (**B**,**E**) and fluorescence intensity (**C**,**F**) were analysed. Bar indicates 50 μm. Data are presented as the means ± SD. ** *p* < 0.01.

**Figure 9 cells-10-01898-f009:**
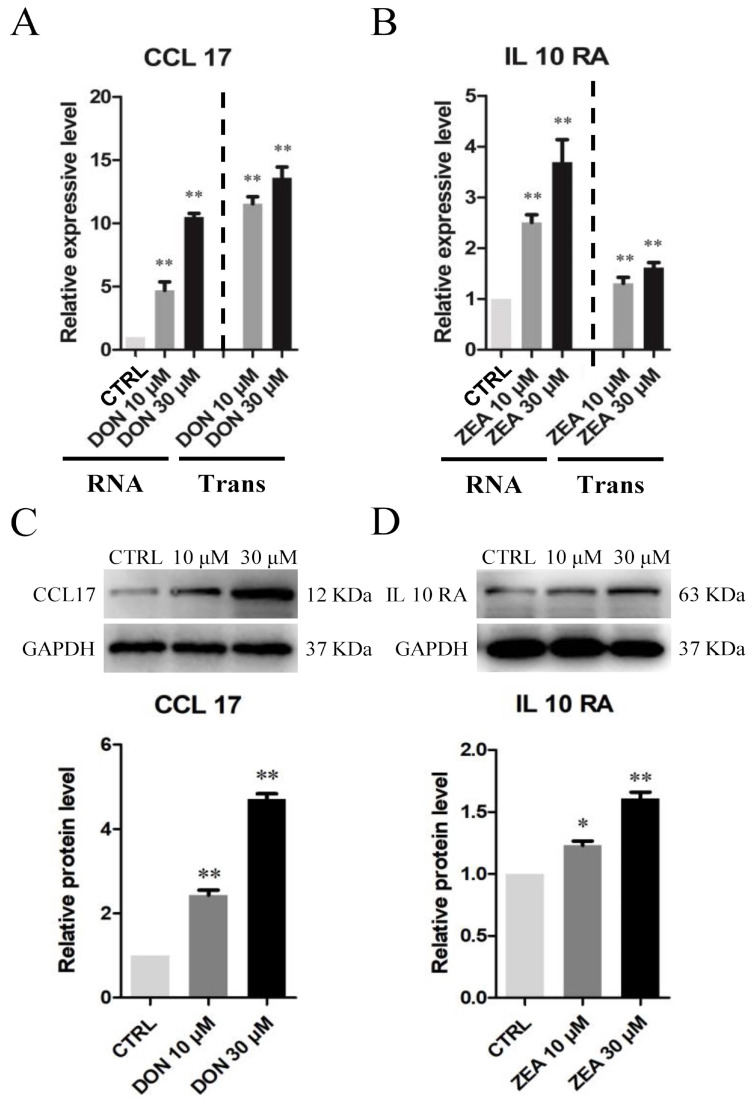
DON and ZEA exposure affects the mRNA and protein abundance levels of inflammation-related genes in cultured *Equus asinus* SCs. (**A**) Quantitative RT-PCR for the *CCL17* transcription factors. (**B**) Quantitative RT-PCR for the *IL10RA* transcription factors. The mRNA levels of the gene were normalised to the *GAPDH* gene. (**C**) Protein levels of *CCL17*/*GAPDH* by Western blot. (**D**) Protein levels of *IL10RA*/*GAPDH* by Western blot. The protein levels were normalised to *GAPDH*. The exposure time was 50 s. The results are presented as the mean ± SD. All experiments were repeated at least three times. * *p* < 0.05; ** *p* < 0.01.

**Figure 10 cells-10-01898-f010:**
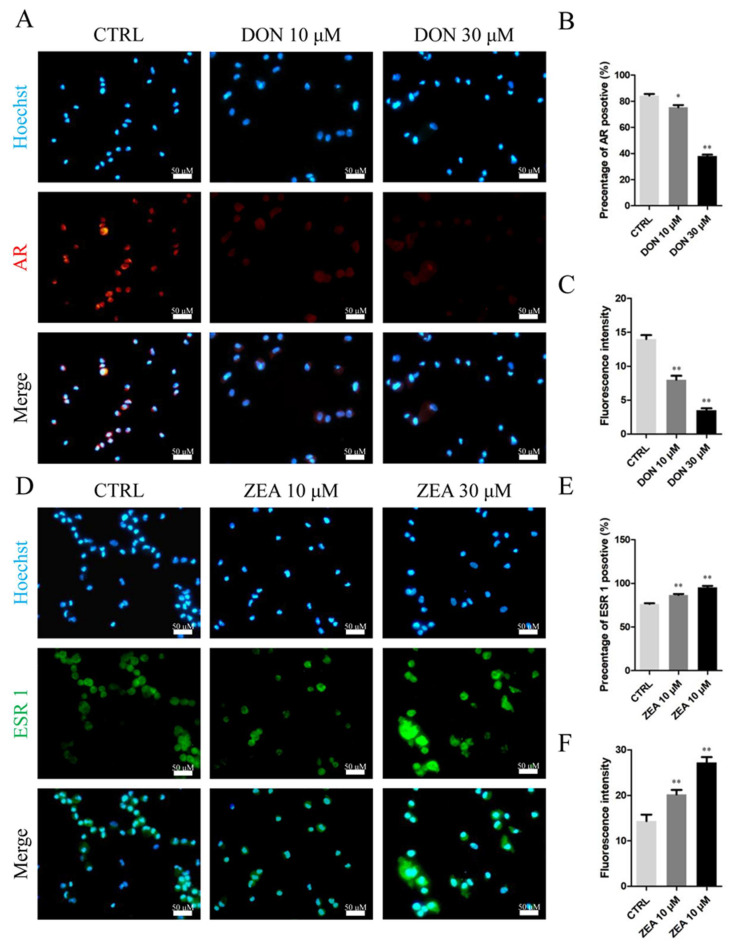
Immunofluorescence assay probing the expression of phosphor-*AR* (**A**) and *ESR1* (**D**) proteins in SCs in the DON and ZEA treatment groups. The percentages of positive cells (**B**,**E**) and fluorescence intensity (**C**,**F**) were analysed. Bar indicates 50 μm. Data are presented as the means ± SD. * *p* < 0.05; ** *p* < 0.01.

**Figure 11 cells-10-01898-f011:**
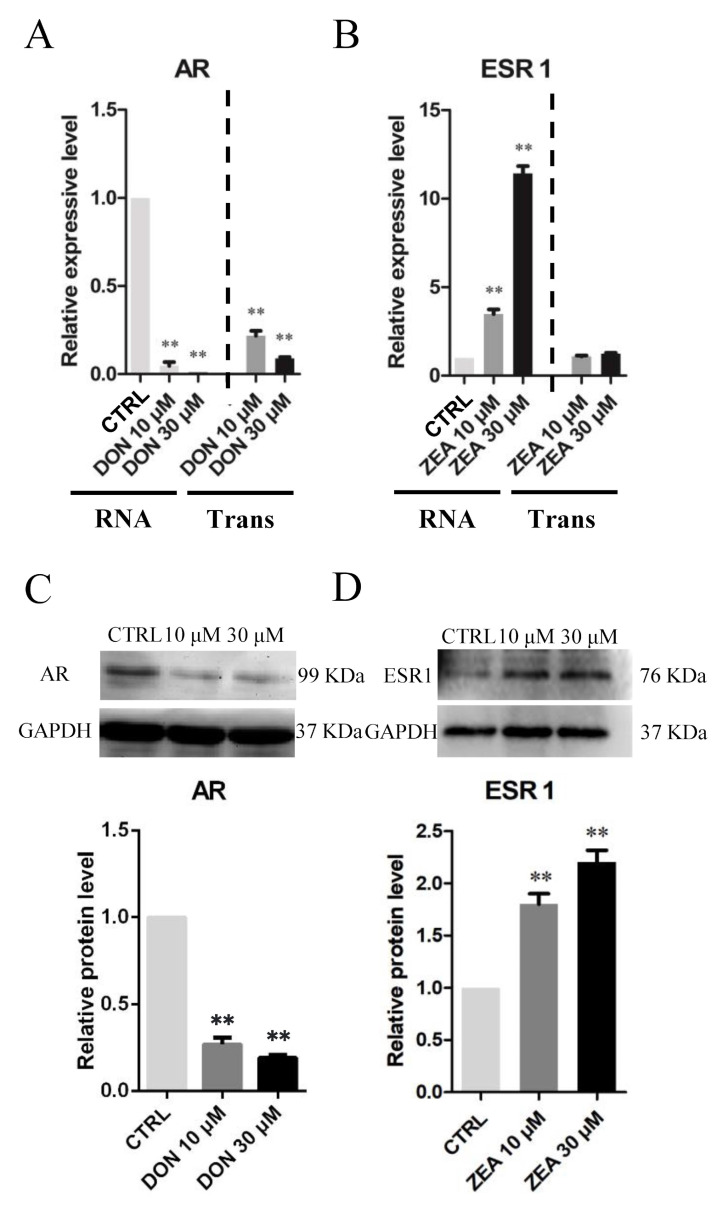
DON and ZEA exposure affects the mRNA and protein abundance of endocrine-related genes in cultured *Equus asinus* SCs. (**A**) Quantitative RT-PCR for the *AR* transcription factors. (**B**) Quantitative RT-PCR for the *ESR* transcription factors. The mRNA levels of the genes were normalised to the *GAPDH* gene. (**C**) Protein levels of *AR*/*GAPDH* by Western blot. (**D**) Protein levels of *ESR*/*GAPDH* by Western blot. The protein levels were normalised to *GAPDH*. The exposure time was 50 s. The results are presented as the means ± SD. All experiments were repeated at least three times. ** *p* < 0.01.

**Figure 12 cells-10-01898-f012:**
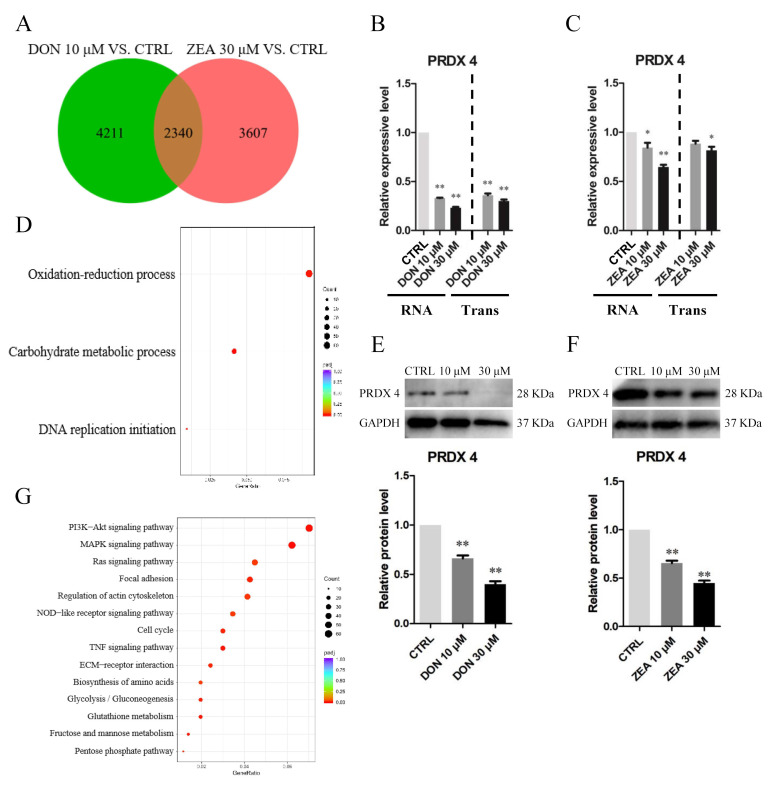
Gene expression of RNA-seq analysis and protein verification between the 10 μM DON- and 30 μM ZEA treated groups. (**A**) Venn diagram showing DEGs in SCs exposed to 10 μM DON and 30 μM ZEA. (**B**,**C**) Quantitative RT-PCR for the *PRDX4* transcription factors in SCs treated with DON and ZEA. (**D**,**G**) GO and KEGG enrichment of 2340 DEGs in *Equus asinus* SCs exposed to 10 μM DON and 30 μM ZEA. (**E**,**F**) The protein levels of *PRDX4*/*GAPDH* by Western blot in DON and ZEA treatment groups. The protein levels were normalised to *GAPDH*. The exposure time was 50 s. The results are presented as the means ± SD. All experiments were repeated at least three times. * *p* < 0.05; ** *p* < 0.01.

**Figure 13 cells-10-01898-f013:**
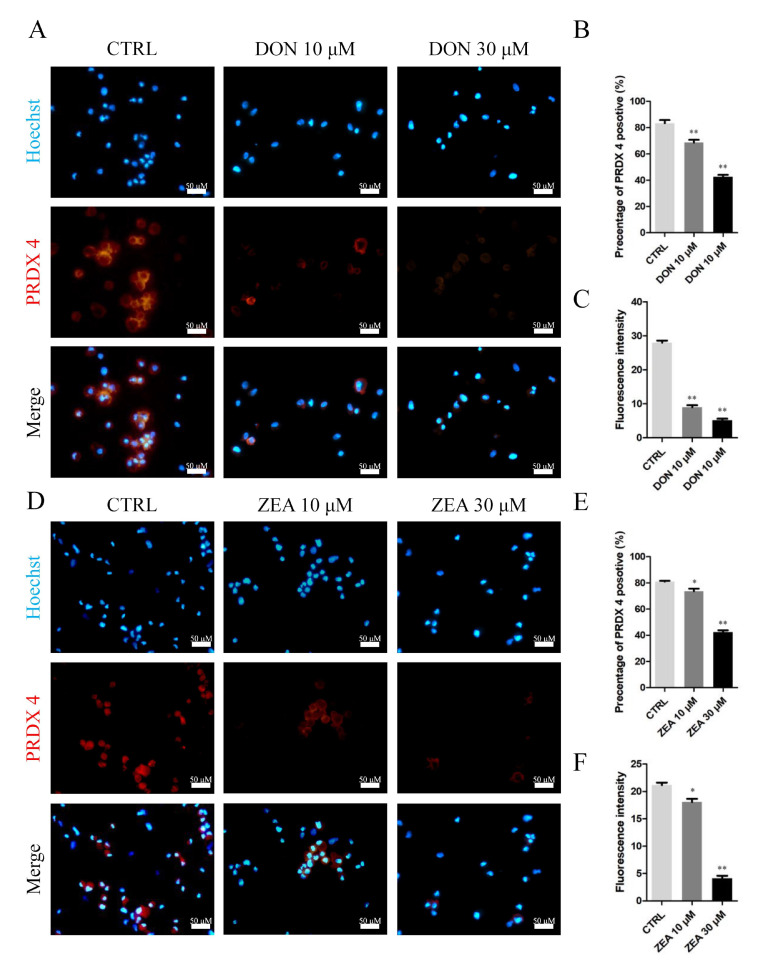
Immunofluorescence assay probing the expression of SC phosphor-*PRDX4* proteins in the DON (**A**) and ZEA (**D**) treatment groups. The percentages of positive cells (**B**,**E**) and fluorescence intensity (**C**,**F**) were analysed. Bar indicates 50 μm. Data are presented as the means ± SD. * *p* < 0.05; ** *p* < 0.01.

**Figure 14 cells-10-01898-f014:**
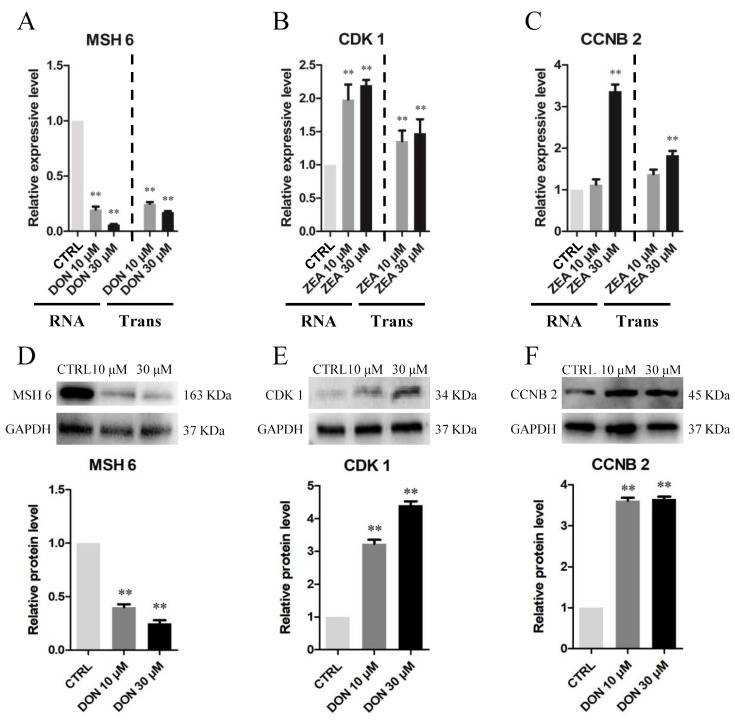
DON and ZEA exposure affected the mRNA and protein abundance levels of tumorigenesis-related genes in cultured GCs. (**A**–**C**) Quantitative RT-PCR for *MSH6* in the DON treatment group, *CDK1* and *CCNB2* in the ZEA exposure groups. The mRNA levels of all genes were normalised to the *GAPDH* gene in SCs. (**D**–**F**) The protein levels of *MSH6*/*GAPDH* in the DON treatment group, *CDK1*/*GAPDH* and *CCNB2*/*GAPDH* in the ZEA treatment group were detected by Western blot. The protein levels were normalised to *GAPDH*. The results are presented as the means ± SD. All experiments were repeated at least three times. ** *p* < 0.01.

**Table 1 cells-10-01898-t001:** Primary antibodies.

Proteins	Article Number	Producer	Origin
GAPDH	AC001	ABclone	Wuhan, China
GSDMD-N	A20197	ABclone	Wuhan, China
Caspase-1	A0964	ABclone	Wuhan, China
CCL 17	A2854	ABclone	Wuhan, China
IL 10 RA	A1830	ABclone	Wuhan, China
ESR 1	A1668	ABclone	Wuhan, China
PRDX 4	A1486	ABclone	Wuhan, China
SOX 9	A2479	ABclone	Wuhan, China
MSH 6	A0983	ABclone	Wuhan, China
CDK 1	A0220	ABclone	Wuhan, China
CCNB 2	A3351	ABclone	Wuhan, China
AR	A2053	ABclone	Wuhan, China
NOX 1	A12309	ABclone	Wuhan, China

**Table 2 cells-10-01898-t002:** Primers used for quantitative-PCR.

Genes	Sequences of Primers (5′ to 3′)	Products	Genbank
*GAPDH*	F: GAAAGCTGCCAAATACGATGAG	136	XM_014866500.1
	R: GAAGGTGGAAGAGTGGATGTC		
*Caspase-1*	F: GGGCACGGGTACAGTAAATAG	114	XM_014851328.1
	R: CGGGCCTTATCCATAACTGTAG		
*GSDMD*	F: GTTATTGGCTCTGACTGGGAC	148	XM_014858171.1
	R: TGAATCCTGACACGCTCTTG		
*CCL 17*	F: ACTGAAGATGCTGTTCCTGG	106	XM_014827258.1
	R: GAAGTACTCCAGACAGCACTC		
*IL 10 RA*	F: GTGGATGAAGTGACTCTGACAG	139	XM_014855577.1
	R: CTCGGAAGTTAGGGAAGATGC		
*IL 32*	F: CAACTCAAGACACCCTCCC	136	XM_014833201.1
	R: AAGTAGCTCGAAACAGGCG		
*IL 23 A*	F: GATGGCTGTGATCCTGAAGG	113	XM_014843912.1
	R: TCCCCTGTGAAAATGTCTGAG		
*AR*	F: AGAGTTGTGTAAGGCAGTGTC	96	XM_014832024.1
	R: TACATGCAATCTCCCCGAAG		
*AIG 1*	F: AGTGGGAACCAAGAACAAGAG	72	XM_014850699.1
	R: CAGTACGGCTAACATCCAGTC		
*ESR 1*	F: GAAGAGACAAACCAAAGCCAG	83	XM_014827966.1
	R: GCCTCCCCAGTGATGTAATATG		
*ESR 2*	F: AGAGGGAAAATGCGTAGAAGG	143	XM_014853708.1
	R: CAGGGTACATACTGGAGTTGAG		
*MSH 6*	F: TTCTCTGGTGCTTGTGGATG	130	XM_014833021.1
	R:ATGGTAGTGGTAGAAAACAGTG		
*MCM 6*	F: GTGAAGGAGTGGGAGAAAGTG	129	XM_014849526.1
	R: AAAGAGCATCAGAAGGACACC		
*POLD 1*	F: TCCGTCATGTGCCGATTC	123	XM_014830334.1
	R: GTAGACCTTCTCAAACTCCAGC		
*CDK 1*	F: CTTGCCAGAGCTTTTGGAATAC	129	XM_014833411.1
	R: ACCTATACTCCAAATGTCAACCG		
*CCNB 2*	F: AACAGAGTTACAACCAGAGCC	143	XM_014835324.1
	R: GCCAATATTTCCATCTGCACTG		
*PFKM*	F: TCCAACTACCTGAACATCGTG	116	XM_014850046.1
	R: GCGTCTACAATCTCTATGATCCG		
*PRDX 4*	F: CTCTGAATGACCTTCCTGTGG	73	XM_014867883.1
	R: TCGGTGTACTGGAATGCTTG		
*NOX 1*	F: ACTACCGTCTCTTCCTTACCG	142	XM_014859678.1
	R: GCAGAAAACTCATTGTCCCAC		

## Data Availability

Nine libraries from the three groups were sequenced, and 715,715,682 raw reads (GEO accession number: GSE172037) were uploaded to Gene Expression Omnibus (https://www.ncbi.nlm.nih.gov/geo/, accessed on 25 July 2021).

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
