# Peer review of "Deoxynivalenol and Zearalenone: Different Mycotoxins with Different Toxic Effects in the Sertoli Cells of Equus asinus"

_cells, 2021, doi:10.3390/cells10081898_

Round 1
Reviewer 1 Report
This was a very good paper. It was highly technical and would take a highly educated person to read and understand it. I would like to suggest a few additions to help the reader in general understanding.
First, most students in veterinary and medical colleges think of Sertoli cells as the source of testicular tumors in certain animals and humans. In lines 49 and 50, you mention follicular haematomas in horses fed oats contaminated with Zearalenone, but nothing about carcinomas in animals which are related to Sertoli cells.
Second, it is mentioned that the average DON or ZEA in China in 2020 was doubled in the following year (line 53 and 54)... these levels seem extremely high compared to the European standards. The authors should make reference to what is seen in the rest of the world compared to China.
Third. it is worth a thought that if the donkeys originated in China as well as the controls, and had ingested these higher levels of DON and ZEA as part of their diet, then results may have been masked or lower than in "virgin" or European animals and Sertoli cells. This could be resolved with non-China controls.
Fourth, define "String data base" which is known for protein:protein interaction.
Fifth, there are many figures, but I feel that these are necessary to document this work
It is most interesting to see the different results of the two mycotoxins on this unique cell type. This data would be important to anyone doing future genetic work with these two mycotoxins.
Reviewer 2 Report
Introduction. The introduction should be improved.
Lines 31-32 should be deleted because they are redundant.
Lines 49-50. The following article should be cited:
Minervini F, Giannoccaro A, Fornelli F, Dell'Aquila ME, Minoia P, Visconti A. Influence of mycotoxin zearalenone and its derivatives (alpha and beta zearalenol) on apoptosis and proliferation of cultured granulosa cells from equine ovaries. Reprod Biol Endocrinol. 2006 Nov 30;4:62. doi: 10.1186/1477-7827-4-62.
The following terms “in vivo”, “in vitro”, “Fusarium”, "Equus asinus" should be written in Italic throughout the text.
Results. The quality of Fig. 4 must be improve because it’s impossible to read. Is the significance level indicated in the graphs?
Discussion. More studies on the effects and DON and ZEA on Sertoli cells (even if conducted in other species) should be reported in the discussion and the results should be compared with those of the present study. For example:
ZHU Lei, JIA Beiping, CAO Li, XU Jingru, ZHAO Jie, FENG Shibin, LI Yu, WU Jinjie, WANG Xichun. Effect of sertoli cells apoptosis in piglets induced by single or combined administration of zearalenone and deoxynivalenol. 2020, Vol. 32 ›› Issue (11): 1954-1962.DOI: 10.3969/j.issn.1004-1524.2020.11.04
Cao L, Zhao J, Xu J, Zhu L, Rahman SU, Feng S, Li Y, Wu J, Wang X. N-acetylcysteine ameliorate cytotoxic injury in piglets sertoli cells induced by zearalenone and deoxynivalenol. Environ Sci Pollut Res Int. 2021 Jun 22. doi: 10.1007/s11356-021-14052-9.
Reviewer 3 Report
The manuscript presented is well written and has a presentation in the line of a Scientific study. The images/graphics are also well presented.
I might suggest to put some of the graphics as Supplementary data to avoid duplication of the information. For example for Figure 3A, 3C and 3E. In this last one it will be easier to see the clusters.
Another suggestion is to give some lines about the possibility of having the mixture of this two mycotoxins and if there is something known regarding to that mixture.
Round 2
Reviewer 2 Report
Lines 60-63. In the sentence the verb is missing.
There is still some words that should be written in Italic throughout the text (for example Fusarium lines 440-443).
I think the quality of Fig. 4 must be higher even if it has been improved.
